# Nanomaterials-Based Colorimetric Immunoassays

**DOI:** 10.3390/nano9030316

**Published:** 2019-02-27

**Authors:** Lin Liu, Yuanqiang Hao, Dehua Deng, Ning Xia

**Affiliations:** 1Henan Province of Key Laboratory of New Optoelectronic Functional Materials, Anyang Normal University, Anyang 455000, China; ddh@aynu.edu.cn; 2Henan Key Laboratory of Biomolecular Recognition and Sensing, College of Chemistry and Chemical Engineering, Shangqiu Normal University, Shangqiu 476000, China; haoyuanqiang@aliyun.com

**Keywords:** colorimetric immunoassays, nanozymes, nanoparticle aggregation, metal ions, nanomaterials

## Abstract

Colorimetric immunoassays for tumor marker detection have attracted considerable attention due to their simplicity and high efficiency. With the achievements of nanotechnology and nanoscience, nanomaterials-based colorimetric immunoassays have been demonstrated to be promising alternatives to conventional colorimetric enzyme-linked immunoassays. This review is focused on the progress in colorimetric immunoassays with the signal amplification of nanomaterials, including nanomaterials-based artificial enzymes to catalyze the chromogenic reactions, analyte-induced aggregation or size/morphology change of nanomaterials, nanomaterials as the carriers for loading enzyme labels, and chromogenic reactions induced by the constituent elements released from nanomaterials.

## 1. Introduction

Molecular diagnostics play an increasing important role in the prevention, identification, and treatment of various diseases in the initial stages. To achieve this goal, many efforts have been devoted to exploring novel and powerful diagnostic tools or methods for the in vitro detection of biomarkers at trace levels, such as polymerase chain reaction and mass spectrometry. Although these methods have adequate sensitivity, they are destructive and often suffer from time-consuming derivatization, high cost, and professional operation. Immunoassays based on the use of antibodies as biorecognition elements have been considered as effective analytical methods for clinical diagnoses [1,2]. The specific recognition and the high equilibrium association constant (10^10^ M^−1^ or even higher) between antibody–antigen interactions endow immunoassays with a high selectivity and sensitivity. Moreover, immunoassays can meet the demand of achieving a cost-effective, on-site, and timely diagnosis.

Immunoassays can be divided into electrochemical, colorimetric, or optical immunoassays depending on the types of signal transducers. Among them, colorimetric immunoassays for tumor marker detection have attracted considerable attention due to their simplicity and high efficiency [3,4]. As one of the conventional colorimetric immunoassays, the enzyme-linked immunosorbent assay (ELISA) normally employs an enzyme-conjugated antibody to directly or indirectly catalyze a chromogenic reaction for quantification, which has been a standard method for diagnosing many diseases. However, the traditional ELISA still has many shortcomings, such as a limited sensitivity, complex operation, high cost, and high reagent consumption [4]. Another popular immunoassay method, transverse flow analysis (LFA), is also widely used as another traditional colorimetric immunoassay method. Gold nanoparticles (AuNPs)-related LFA is an early example of a nanomaterial-related colorimetric immunoassay. Because LFA is very simple in operation and reading, it provides obvious advantages in point-of-care testing. However, the sensitivity of LFA is usually not comparable to that of ELISA [5].

With the achievements of nanotechnology and nanoscience, the stability improvement and signal amplification in conventional colorimetric immunoassays is achieved by taking advantage of the unique property of nanomaterials [6,7]. For instance, scientists can introduce peroxidase-like nanomaterials into immunoassays for the replacement of natural enzymes to overcome their shortcomings, such as a low operational stability, the dependence of external conditions (e.g., temperature and pH value), and difficulties in preparation, purification, and storage. Based on the behavior of nanomaterials, the strategies of colorimetric immunoassays can be summarized as four types: (1) Nanomaterial-based artificial enzymes, which can catalyze the chromogenic reactions; (2) analyte-induced aggregation or size/morphology transition of nanoparticles; (3) nanomaterials as the carriers for loading enzyme labels; and (4) chromogenic reactions induced by the constituent elements released from nanomaterials. In this review, we focus on the recent advances in colorimetric immunoassays with various types of nanomaterials and diverse clever methodologies. Nanomaterials, such as polymeric nanoparticles or nanofibers, can enhance the capture efficiency of sensor surfaces due to the large surface-to-volume ratios [8,9,10,11,12,13]. The efficient immobilization of antibodies on the nanoscale surface to enhance the capturing of the target is not summarized in this review. Due to the explosion of academic papers related to this extremely wide research field, we may undoubtedly miss many important findings. We sincerely apologize to the authors for their interesting works that are overlooked in this review.

## 2. Natural Enzymes for Colorimetric Immunoassays

Due to the intrinsic and remarkable catalytic properties, enzymes have usually been adapted in biochemical-related assays as signal amplifiers. In colorimetric immunoassays, there are three enzymes that are most frequently used, including horseradish peroxidase (HRP), alkaline phosphatase (ALP), and β-galactosidase, until now. The detection principle is related to the oxidation or reduction of a substrate under the catalysis of a natural enzyme. The change of color or signal intensity is linear with the target concentration, allowing for a naked-eye readout or quantification with a spectrometer, even with a smartphone.

Among the kinds of HRP-catalyzed colorimetric reactions, the HRP/3,3’,5,5’-tetramethylbenzidine (TMB) immunoassay system is commonly used in most of the commercial ELISAs. Through the addition of H_2_O_2_, HRP can catalyze the oxidation of TMB into blue product TMB^2+^ with characteristic absorption peaks at 653 nm. Furthermore, when hydrochloric acid or sulfuric acid is added to terminate the reaction, the solution color will change from blue to yellow with the absorption peak shifting from 653 to 405 nm [14]. Meanwhile, the yellow product can quantitatively etch gold nanorods, displaying vivid colors as colorful as a rainbow [15]. *O*-phenylenediamine (OPD), a colorless and non-fluorescent compound, can be oxidized by H_2_O_2_ to form a yellow and luminescent compound, 2,3-diaminophenazine (DAP), under the catalysis of HRP, which has been used as the sensitive substrate for the designing of colorimetric or fluorescent assays [16,17]. 

Alkaline phosphatase (ALP) is a hydrolytic enzyme, which can regulate the dephosphorylation process and has been identified as an important biomarker of liver dysfunction and bone diseases [18,19]. Importantly, ALP shows an optimal enzyme reaction rate in catalyzing the dephosphorylation or transphosphorylation of phosphate molecules in an alkaline environment and has been employed as a signal-amplification enzyme in colorimetric immunoassays, especially in ELISA. Typically, ALP can hydrolyze *p*-nitrophenyl phosphate (*p*-NPP) into the yellow colored *p*-nitrophenol (*p*-NP) [20]. Besides, ALP can also catalyze the conversion of 4-aminophenyl phosphate (4-APP) or ascorbic acid-phosphate (AA-P) into reductant 4-aminophenol (4-AP) or ascorbic acid (AA). The enzymatic product can interact with metal salts or organic molecules to produce chromogenic species [21,22,23,24]. For example, Chen’s group demonstrated that the generated AA can catalyze the reduction of colorless tris(bathophenanthroline) iron(III) into pink red tris-(bathophenanthroline) iron(II) by tris(2-carboxyethyl)phosphine [21]. The signal was amplified by the so-called redox cycling between tris(bathophenanthroline) iron(III), AA, and tris(2-chloroethyl) phosphate (TCEP).

Glucose oxidase (GOx) can catalyze the oxidation of glucose to gluconic acid and H_2_O_2_ in the presence of dissolved O_2_. Unlike HRP and ALP, although GOx has some advantages, such as a low cost and high stability, it is hardly used in colorimetric immunoassays due to the lack of a good chromogenic substrate [25]. However, GOx can be coupled with other enzymes (such as HRP) or a colorimetric reaction to carry out the GOx-catalyzed cascade amplification to broaden the application of GOx, in which the product of H_2_O_2_ is the substrate of the next enzyme-catalyzed or chromogenic reaction [26,27]. For example, Yang’s group suggested that H_2_O_2_ (the product of GOx-catalyzed oxidization of glucose) can oxidize Fe(II) into Fe(III). The resulting Fe(III) can rapidly coordinate with squaric acid, leading to a change of the solution color from bluish purple to bluish red [28]. The H_2_O_2_-induced iodine-starch complex and Fenton reaction-induced TMB oxidation are also integrated in colorimetric immunoassays [29,30]. Recently, GOx has been employed to develop colorimetric immunoassays based on GOx-catalyzed growth, aggregation, and surface changes of nanomaterials [31,32,33].

Some chemicals (e.g., H_2_O_2_ and AA) can reduce metal salts into nanoparticles or conversely oxidize nanoparticles into metal salts (e.g., MnO_2_ nanoparticles to Mn^2+^) under certain conditions, resulting in a change of the solution color. Such reactions allow for the application of some native enzymes in colorimetric immunoassays. For example, catalase can decompose H_2_O_2_ into H_2_O and O_2_ [34]. AA can be oxidized to inactive dehydroascorbic acid by ascorbate oxidase [35]. β-galactosidase (β-gal) can hydrolyze the substrate of chlorophenol red-β-D-galactopyranoside into a red-violet product of chlorophenol red. Henry’s group designed colorimetric paper-based analytical devices for the diagnosis of foodborne diseases based on the β-gal-catalyzed reaction [36,37]. More related works and elaborate methods based on these enzymes will be discussed in the following sections.

## 3. Nanomaterials-Based Artificial Enzymes

Nanomaterials-based artificial enzymes, also called nanozymes, were confined to nanomaterials with enzyme-mimicking activity. In contrast to natural enzymes and conventional artificial enzymes, nanozymes possess various distinguish advantages, such as a robustness to harsh environments, tunable catalytic activities, low cost, and ease of mass production while retaining the inherent unique properties of nanomaterials. Recently, nanozymes have exhibited huge potential in the applications of biosensing, immunoassays, therapeutics, catalysis, and environmental protection. Thanks to the rapid development of nanotechnology and the distinguished properties of nanomaterials, numerous nanomaterials with different compositions and various nanostructures have been successfully synthesized and demonstrated to possess enzyme-like activities, mainly including peroxidase-, oxidase-, catalase-, and super-oxide dismutase-like nanozymes. For clarity, the nanozymes are classified into noble metal, carbon, metal-oxide, and metal-organic frameworks according to the institute elements and structural characteristics of the nanomaterials.

### 3.1. Noble Metal-Based Nanozymes

#### 3.1.1. Gold

Gold was regarded as being chemically inert at one time. It was surprising when Rossi et al. found “hidden talents” of gold nanoparticles (AuNPs) in that citrate-coated AuNPs exhibited GOx mimicking activity [38]. Further investigations demonstrated that the catalytic properties of AuNPs are related to the composition, size, charge, and the configuration of the coating agents on the surface of the nanoparticles [39,40,41,42,43]. For example, Huang’s group found that Hg^2+^ and other metal ions (such as Ag^+^ and Pb^2+^) can relatively enhance the peroxidase mimetic activities of AuNPs [39,44,45]. They prepared AuNPs/GO hybrids with tannic acid (TA) as both the reducing and stabilizing agent and demonstrated the applications of the Hg^2+^-stimulated peroxidase-like activity of AuNPs/GO hybrids for respiratory syncytial virus (RSV) detection with a sandwich-like colorimetric immunoassay [46]. Lee’s group reported a simple MagNBs-based nano(e)zyme-linked immunosorbent assay (MagLISA) utilizing MagNBs for biomarker enrichment and the peroxidase nanozymes of AuNPs for signal amplification (Figure 1A) [47]. After the immunoreaction and magnetic separation, the positively charged AuNPs accelerated the oxidation reaction of TMB by H_2_O_2_. The colorimetric signal is directly correlated with the concentration of influenza virus. The limit of detection (LOD) of this MagLISA was greatly improved (10^−14^ g/mL). Additionally, AuNPs can also catalyze the reduction of *p*-nitrophenol to *p*-aminophenol, leading to a significant color change from yellow to colorless. Qu et al. developed a colorimetric immunosensor for the determination of carbohydrate antigen 125 using hollow polydopamine-AuNPs as the signal labels [48]. Among the big family of gold nanomaterials, gold nanoclusters (AuNCs) consisting of several to tens of atoms also have peroxidase-like activity [40,49]. Chen and co-workers developed a plasmonic immunoassay to enable visually quantitative determination of ultratrace targets using peroxidase-mimetic AuNCs (Figure 1B) [50]. AuNCs conjugated on the outer layer antibody can catalyze the decomposition of H_2_O_2_, thus reducing HAuCl_4_ into AuNPs for the colorimetric signal output. Like AuNPs, the peroxidase-like properties can also be inhibited by metal ions and small molecules [49,51]. Thus, Zhang’s group reported a competitive ELISA for the determination of dibutyl phthalate (DBP) based on the inhibition of the catalysis of bovine serum albumin (BSA)-capped AuNCs triggered by dissolved Ag(I) released from the second antibody-labeled AgNPs [52]. Gold nanozymes exhibit a high catalytic activity; however, modifications of the gold surface with antibodies depresses their catalytic activity due to the hindrance effect. To overcome this bottleneck and improve the detection sensitivity of nanozyme-based immunoassays, Khoris et al. developed a new detection strategy in which a silver-coated gold core/shell nanohybrid structure was generated by the in-situ growth of silver on the gold surface [53]. The resulting Au/Ag nanohybrid exhibited enhanced peroxidase-like activity for the oxidation of TMB by H_2_O_2_. For the colorimetric immunoassay of norovirus, the strategy with Au/Ag nanohybrid as the signal label exhibited 1000- and 100-fold higher sensitivity compared to the gold immunoassay and horseradish peroxidase (HRP)-based ELISA, respectively.

#### 3.1.2. Palladium

Xia’s group found that a coating of ultrathin Ir shell (a few atomic layers) on palladium nanoparticles (Pd NCs) significantly enhanced the peroxidase mimic efficiency of the initial Pd NCs toward TMB oxidation (Figure 2A). The resulting Pd-Ir NCs were used to develop a colorimetric ELISA for prostate surface antigen (PSA) detection with a much lower LOD than the conventional ELISA (Figure 2B) [54]. Recently, they also encapsulated Pd-Ir NCs within gold vesicles (GVs) as the signal elements for the colorimetric assay of human PSA with a greatly improved sensitivity (Figure 2C) [55]. In this enzyme-free signal amplification strategy, GVs captured by the antigen liberated thousands of individual Pd-Ir NPs at elevated temperatures due to the heat-induced breakup of the GV membrane. The released Pd-Ir NPs caused an intense color signal through the catalyzed oxidation of TMB by H_2_O_2_.

#### 3.1.3. Platinum

Besides Pd-Ir NCs, poly-(vinylpyrrolidone) (PVP)-capped Pt nanocubes also possess excellent peroxidase-like activities toward TMB oxidation. However, the activity can be specifically and strongly inactivated by Ag(I) within several minutes by binding to the surface of Pt nanocubes (Figure 3A) [56,57,58]. For this view, Xia and co-workers reported a non-enzyme colorimetric immunoassay for PSA detection based on the antibody-conjugated silver nanoparticles (Ag NPs) and the PVP-capped Pt nanocubes [59]. In this work, Ag NPs were chemically etched by H_2_O_2_, and the product, Ag(I), was deposited on the surface of the Pt nanocubes, largely eliminating the generation of the colored product. More importantly, the sensitivity of the system could be reduced to 0.031 pg/mL by the coupling with the “silver enhancement” technique. Furthermore, Xia’s group also decorated conventional AuNPs with conformal, thin skins of Pt to form unique Au@Pt core@shell NPs, in which the plasmonic property of the AuNP is well retained and the Pt shells endow the Au@Pt NPs with ultrahigh peroxidase-like catalytic activity (Figure 3B) [60]. The dual functional Au@Pt NPs offered two different detection models to achieve an “on-demand” tuning of the detection performance and were applied to construct a lateral flow assay of the PSA. Recently, Stevens’s group developed a paper-based lateral flow immunoassay (LFIA) for the naked-eye detection of p24 and acute-phase HIV in under 20 min using peroxidase-mimicking porous platinum core-shell nanocatalysts (Figure 3C) [61]. To improve the dispersion stability and preserve the catalytic activity after surface modification with antibody, Wang’s group encapsulated platinum nanoparticles in mesoporous silica (Pt@mSiO_2_ NPs) through the sol-gel method. The peroxidase-like nanocomposites were further applied for the colorimetric detection of human choionic gonadotophin (hCG).

### 3.2. Metal Oxide-Based Nanozymes

#### 3.2.1. Magnetic Nanoparticles

Magnetic nanoparticles (MNPs) are particularly useful for a wide range of biomedical, environmental, and catalytic applications. They can be conjugated with enzymes, DNA, peptides, or antibodies for further functionalization. In 2007, Yan et al. firstly found that Fe_3_O_4_ MNPs possessed an intrinsic peroxidase-like activity similar to the natural peroxidases (Figure 4A) [62]. They developed two immunosensors using Fe_3_O_4_ MNPs as both the peroxidase and the magnetic separators. In this method, the antibody-modified MNPs provide three functions: Capture, separation, and detection. After that, various colorimetric immunoassays were developed based on the unique HRP-like property of MNPs. For instance, Gao’s group employed the catalytic and magnetic property of chitosan-modified MNPs to develop a magnetic nanoparticle-linked immunosorbent assay for the detection of mouse immunoglobulin G (IgG) and tumor marker carcinoembryonic antigen (CEA) [63]. Tang and co-workers reported a novel reverse colorimetric immunoassay for the sensitive detection of low-abundance protein based on the highly catalytic efficient catalase carried by AuNPs and the peroxidase-like activity of magnetic beads (Figure 4B) [64]. The magnetic beads and AuNPs were functionalized with an anti-PSA capture antibody and catalase/anti-PSA detection antibody, respectively. A specific sandwich-type immunoassay was formed in the presence of PSA. More PSA would provide more catalase. The catalase caused the consumption of H_2_O_2_ in the detection solution, leading to a partially reduced catalytic efficiency of the magnetic beads as their peroxidase mimics the TMB/H_2_O_2_ system.

Although MNPs have attracted worldly attention due to their superior characteristics, their relatively low activity stemming solely from the presence of superficial ferrous atoms has hindered their widespread applications in practice. To overcome this inherent limitation, scientists have made great efforts to enhance the catalytic activity of free MNPs, by integrating enzymes or nanozymes into MNPs [66]. For example, Park’s group incorporated MNPs and Pt NPs into ordered mesoporous carbon and found that the nanocomposites have 50 times higher catalytic efficiency than that of the free MNPs (Figure 4C) [65]. Next, the surface of the nanocomposite was treated with acid to produce carboxylic residues for further conjugation of antibody. The resulting conjugates were used to perform colorimetric immunoassay for rapid, robust, and convenient detection (within 3 min) of clinically important target pathogens. Moreover, Gu and co-workers presented a simple method to prepare γ-Fe_2_O_3_ nanoparticles modified by Prussian blue (PB) at different levels and found that their peroxidase-like activity was enhanced with the increasing PB proportion [67]. They further conjugated staphylococcal protein A (SPA) to the surface of the nanoparticles to construct an enzyme immunoassay for the detection of IgG.

#### 3.2.2. Cerium Oxide Nanoparticles

Cerium oxide nanoparticle (CeO_2_ NP, nanoceria) has been identified as a good catalyst in various applications, such as biomedical and catalytic applications, due to the presence of oxygen vacancies and a mixed valence state (Ce^3+^, Ce^4+^). According to the published reports, CeO_2_ NPs have been demonstrated to possess multienzyme, such as superoxide dismutase (SOD), catalase, oxidase, and phosphatase mimetic properties [68]. Perez and co-workers first reported that nanoceria had an intrinsic oxidase-like activity to organic dyes and small molecules at an acidic pH without any oxidizing agents (e.g., H_2_O_2_) [69]. They also investigated the relationship between the oxidase activity and pH, the size of nanoceria as well as the thickness of the polymer coating. In their work, poly(acrylic acid)-coated nanoceria was conjugated with folic acid and used to develop a faster and cheaper immunoassay for the lung cancer cell line (A-549). Next, Perez and co-workers demonstrated that the oxidase mimetic ability of poly(acrylic acid)-coated nanoceria could be well tuned by changing the solution pH (Figure 5) [70]. Nanoceria can partially oxidize the nonfluorescent substrate, ampliflu, to the very stable fluorescent product, resorufin, at pH 6–8 due to the weak oxidase-like activity. However, the ampliflu was completely oxidized into the nonfluorescent product, resazurin, at acidic pH (4–5). The tunable nanoceria-mediated oxidation of ampliflu was thus used to develop a sensitive cell-based ELISA at neutral without the use of H_2_O_2_. Because of a large surface area and a high surface Ce^3+^ fraction, porous nanorods of ceria (PN-Ceria) have stronger intrinsic peroxidase activity than that of other nanoceria and have been applied for the design of a colorimetric immunosensor for breast cancer diagnosis with a high sensitivity and a low LOD [71]. Based on the excellent oxidase activity, nanoceria, combined with carboxyl-functionalized Fe_3_O_4_ NPs, was used to develop a magnetic colorimetric immunoassay for the sensitive detection of human interleukin-6 (IL-6) [72]. Nanoceria catalyzed the oxidation of *o*-phenylenediamine to a yellow product, 2,3-diaminophenazine, without the presence of H_2_O_2_.

#### 3.2.3. Other Metal Oxides’ Nanomaterials

Lv and co-workers reported that BSA-stabilized MnO_2_ NPs exhibited highly peroxidase-, oxidase-, and catalase-like activities [73]. Based on the findings, BSA-MnO_2_ NPs were used as the immunoassay labels for the colorimetric detection of goat anti-human IgG in place of HRP. Furthermore, Zhang et al. prepared five different MnO_2_ nanomaterials and investigated the catalytic abilities and mechanisms of these MnO_2_ nanomaterials [74]. They found that MnO_2_ nanowires are more stable at room temperature and have the highest oxidase-like catalytic property. Finally, a sensitive and selective MnO_2_ nanowires-mediated immunoassay was carried out for the analysis of the pathogen. Additionally, Xie’s group developed an immunosensor for the detection of α-fetoprotein (AFP) based on the high peroxidase-like catalytic activity of MnO_2_ NPs and the tyramine-triggered signal amplification (Figure 6A) [75]. Li’s group reported the rapid colorimetric immunoassay of *V. parahaemolyticus* with the combination of a magnetic bead-based sandwich immunoassay and signal amplification using oxidase mimics of MnO_2_ NPs (Figure 6B) [76]. When being reduced to Mn^2+^ by reducing substances, MnO_2_ NPs lost their peroxidase-like activity or oxidase-like activity. For this view, Tang’s group developed an ascorbate oxidase-based cascade amplification for an immunoassay of Aflatoxin B1 (AFB_1_) (Figure 6C). The catalytic oxidation of AA into dehydroascorbic acid hampered the MnO_2_ NPs-catalyzed oxidation of TMB [35,77].

Polyoxometalates (POMs) are well-known metal-oxo-cluster compounds for the fabrication of organic-inorganic hybrids. Wang’s group prepared the hybrid NPs through the self-assembly of phosphovanadomolybdate (PVM) and folate (FA) [78]. They found that PVM-FA hybrid NPs have unique oxidase-like activities, which can quickly oxidize TMB at acidic and neutral pH. The oxidase-like mimetic NPs ware used to construct colorimetric multiplexed immunoassays for folate receptor-overexpressing tumor cell in the pH range of 3–7 without pretreatment. Furthermore, folate-functionalized iron-substituted POM FA-γ-[(FeOH_2_)_2_SiW_10_O_36_] (FA-Fe_2_SiW_10_) inorganic-organic hybrid NPs were synthesized by Wang and co-workers. The hybrid NPs were applied in the colorimetric multiplexed immunoassays of cancer cells [79].

Nanozymes-based immunoassays are accurate and effective. However, there are still some drawbacks in practical applications, such as the introduction of H_2_O_2_. To explore more stable and efficient immunoassays, some photocatalysts used in the fields of energy and the environment were integrated with the immunoassay with the aid of sunlight. TiO_2_ is a well-known wide band gap semiconductor with an ultraviolet absorption band. Wang’s group reported that catechol (CA) can bind on the surface of TiO_2_ NPs (TiO_2_-CA NPs) via the specificity and high affinity of diol ligands to Ti(IV). The inert TiO_2_ NPs were thus activated to exhibit highly efficient oxidase mimicking activity for the catalysis of the oxidation of TMB under visible light (λ ≥ 400 nm) irradiation, utilizing dissolved oxygen as an electron acceptor (Figure 7A) [80]. Based on the dual reaction between ALP and TiO_2_-CA NPs, a novel immunoassay for the determination of mouse IgG was developed with a low LOD (2.0 pg/mL), which is 4500-fold lower than that of the standard ELISA kit. Moreover, Wang’s group also found that BSA-Au NCs, GO, and silver halide (AgX, X = Cl, Br, I) NPs exhibited photo-responsive peroxidase-like activities under visible light irradiation [81,82,83]. Typically, with photoactivation, chitosan (CS) modified (CS-AgX) NPs exhibit a higher peroxidase-like activity over the natural peroxidase or other existing peroxidase-like nanomaterials. For this view, they developed a colorimetric immunoassay method for the detection of cancer cells with an LOD that was down to 100 cells (Figure 7B) [83].

### 3.3. Metal-Organic Frameworks-Based Nanozymes

Metal-organic frameworks (MOFs), consisting of metal-containing nodes and organic linkers by strong bonds, have received more and more attention in the past two decades for their ordered nanoporous system, thermal stability, and tunable physicochemical properties. The unique composition, structural diversity, and tunable size make MOFs good alternatives for novel nanozymes [84]. Besides, MOFs can be a better candidate as a host for biomolecule encapsulation, because they can protect the biomolecules from harsh environments and preserve their original biological functions [85,86,87]. For the convenience of this review, we integrated covalent organic frameworks (COFs) and amorphous three-dimensional covalent organic polymer (COP) into this section. Tan et al. reported a “one-pot” method for the fabrication of dual-functional RIgG@Cu–MOF composites for colorimetric immunoassays (Figure 8A) [88]. In the composite, Cu-MOF can provide superior shielding against harsh environments, but has no influence on the original capture ability of RIgG to the corresponding antigen (mIgG). Meanwhile, Cu-MOF can catalyze the oxidation of TMB in the presence of H_2_O_2_ due to its intrinsic peroxidase property. Under optimized conditions, this simple, direct, and less expensive method has a high sensitivity toward mIgG with an LOD of 0.34 ng/mL. As the active center of many natural enzymes, metal ion-containing porphyrin compound has been widely utilized as an artificial enzyme mimic for heterogeneous catalysis to be immobilized or encapsulated into various scaffolds, including porous framework materials or other nanomaterials [89,90,91,92]. Zhang et al. reported a novel method to synthesize iron porphyrin-based COP (FePor-TFPA-COP), which possessed a large surface area and abundant surface catalytic active sites and exhibited strong peroxidase-like activity toward TMB oxidation by H_2_O_2_ (Figure 8B) [93]. Using the porous materials as signal labels, they developed a colorimetric immunoassay for AFP detection with high specificity, sensitivity, and stability.

### 3.4. Carbon-Based Nanozymes

As an-atom-thick planar sheet of sp^2^-hybridized carbon atoms, graphene has attracted much attention in materials science and biotechnology. Besides the superior characters of graphene, such as a super fluorescence quenching efficiency, excellent electron conductivity, and good biocompatibility, carboxyl-modified graphene oxide (GO) also possesses peroxidase-like activity for the catalytic oxidation of peroxidase substrate TMB by H_2_O_2_ to trigger a blue color reaction, which was first reported by Qu’s group in 2010 [94]. The proposed mechanism is that electron transfer occurs from the top of the valence band of GO to the lowest unoccupied molecular orbital (LUMO) of H_2_O_2_. Moreover, Yang’s group found that GO can catalyze the oxidation of hydroquinone by H_2_O_2_ to trigger a brown color solution and thus developed a GO-based colorimetric immunoassay for the visual detection of the cancer biomarker of PSA [95]. In the presence of PSA, a sandwich-like immunocomplex is formed between the GO-Ab_2_ and MB-Ab_1_. After the magnetic separation, the GO-Ab_2_ suspension at a certain content was mixed with hydroquinone and H_2_O_2_ to result in the appearance of a brown color. This method can be used to determine PSA in real samples. 

Carbon dots (CDs), including carbon quantum dots, graphene quantum dots (GQDs), and polymer carbon dots (PCDs), have aroused considerable attention because of their promising advantages [96]. In contrast to the conventional organic dyes and semiconductor quantum dots, CDs exhibit a series of advantages, including chemical inertness, ease of preparation, tunable luminescence properties (fluorescence and phosphorescence), ideal electro- or photo-catalytic features, low cytotoxicity, and excellent biocompatibility. Recently, Huang’s and Qu’s group have demonstrated that CDs exhibit strong intrinsic peroxidase-like activity [97,98]. Further investigations indicated that the metal-doped carbon quantum dots have an enhanced peroxidase-like activity and the GQDs exhibit higher peroxidase-like activity than GO [99,100,101]. However, there are few works focusing on the development of colorimetric immunoassays based on the peroxidase-like CDs. Recently, Wei’s group prepared iron and nitrogen co-doped CDs (Fe-N-CDs) derived from *L*-tartaric acid, urea, and FeCl_3_ through the one-step microwave synthesis method [99]. They found that the resulting Fe-N-CDs showed superior catalytic performances for the oxidization of TMB by H_2_O_2_. Using antibody-conjugated Fe-N-CDs as the labels of colorimetric immunoassay, CEA was detected with an LOD down to 0.1 pg/mL.

## 4. Nanomaterials as Colorimetric Substrates

In the above colorimetric immunoassays, the visible signal is usually generated through the conversion of catalytic substrates into chromogenic molecules. The intensity of the solution color is strictly related to the extinction coefficient of the chromogenic molecules. Thus, the color change may not be observed in the presence of a low concentration of analyte due to the low extinction coefficient of the substrates or products, limiting the applications of immunoassays for ultrasensitive and naked-eye detection.

Plasmonics is a unique phenomenon caused by the collective oscillation of conduction band electrons onto the metal surface with light stimulation. Unlike a bulk metal or an extended metal surface, if the surface electrons of NPs were excited by the resonance light with a longer wavelength than the size of NPs, the plasmon would be confined to the surface of metallic nanoparticles (such as gold, silver, or alloys), known as localized surface plasmon resonance (LSPR). Meanwhile, there will be a typical LSPR absorption band within the visible or near-infrared frequency range, together with the appearance of a relevant color of the solution of NPs, which relies on the nanoparticles’ size, shape, surface coating, and composition, and the dielectric constant of the environments and the inter-distance between nanoparticles. Therefore, changes in the above properties can induce a shift in the wavelength and broadening of the spectra, accompanyied with a significant change in the color or intensity. Based on the unique plasmonic properties, various plasmonic sensing strategies have been widely applied in the detection of biological and chemical molecules. Interested readers are referred to recent review articles elsewhere [102,103].

By combining immunoassays with LSPR-based colorimetric assays, plasmonic immunoassays enable the detection of low concentrations of analytes with the naked-eye [104]. Enzymes, employed in antibody-conjugated forms, can convert substrates into triggers, resulting in the aggregation or enlargement of NPs and a change of the color of the NPs solution. In plasmonic nanoparticle-based colorimetric immunoassays, the strategies of colorimetric immunoassays are mainly based on two aspects: The aggregation or dispersion of nanoparticles and the size/morphology change of the nanoparticle surface.

### 4.1. Nanoparticle Aggregation

The aggregation of plasmon nanoparticles with less than a 2.5 of separation-to-diameter ratio can induce strong inter-particle plasmon coupling, resulting in the LSPR shift and an apparent color transition of the nanoparticle suspension [105]. Small organic molecules with a thiol group or net charge can bind to the metal NPs through an Au-S bond, hydrogen bond, or electrostatic interactions, thus triggering the aggregation of nanoparticles. Boronic acids can react with *cis* diol-containing molecules to form stable complexes, which has been used to develop optical and potentiometric sensors [106,107]. Tseng’s group found that benzene-1,4-diboronic acid (BDBA) can effectively induce the aggregation of citrate-capped AuNPs through the interaction between the α-hydroxycarboxylate of citrate and the boronic acid group of BDBA (Figure 9A) [108]. However, once the boronic group was oxidized into phenol by H_2_O_2_, the aggregation citrate-capped AuNPs was inhibited [109]. Based on this fact, they reported the selective naked-eye detection of rabbit IgG and human PSA with the aid of GOx. Cysteine can induce rapid aggregation of AuNPs by binding to the surface of metal NPs through intermolecular hydrogen bonds or electrostatic interactions between the amine and carboxyl groups [110]. Abbas et al. reported an enzyme-free colorimetric immunoassay with the signal amplification of cysteine-loaded liposomes (Cys-liposome) (Figure 9B) [111]. After the pathogen capture, the immunocomplex is formed and labeled with Cys-liposome through the biotin–streptavidin interaction. Next, the introduction of the surfactant caused the immediate hydrolysis of the liposomes, thus leading to the release of encapsulated cysteine molecules. The released cysteine triggered the aggregation of AuNPs. The approach enabled the naked-eye detection of the target at a concentration below 6.7 attomolar. The value is six orders of magnitude lower than that of the conventional ELISA. Additionally, Chen’s group also reported a colorimetric immunoassay for pathogen detection through the acetylcholinesterase (AChE)-catalyzed hydrolysis reaction. The sensitivity is comparable to that of the polymerase chain reaction (Figure 9C) [112]. In this study, AChE hydrolyzed acetylthiocholine into a sulfhydryl compound (thiocholine), which caused the agglomeration of AuNPs. The detection sensitivity was greatly enhanced due to the signal amplification of AChE-catalyzed hydrolysis and the high density loading of Ab_2_ on the MBs. Jiang’s group also reported an ultrasensitive plasmonic immunoassay for the determination of total antibodies to Treponema pallidum by the AChE-catalyzed hydrolysis of acetylthiocholine [113]. Additionally, iodide can catalyze the oxidation of thiol molecules (for example, cysteine and glutathione) into disulfide molecules (such as cystine and glutathione disulfide) by H_2_O_2_. The resulting disulfides exhibit a poor ability to trigger the aggregation of NPs. Based on this fact, Jiang’s group developed plasmonic immunoassays based on HRP-catalyzed oxidation of iodide and iodide-catalyzed oxidation of cysteine to modulate the state of AuNPs (Figure 9D) [114]. Cu^2+^ can catalyze the oxidation of cysteine into cystine by O_2_, thus depressing cysteine-induced aggregation of AuNPs. This method allowed for the detection of Cu^2+^ with an LOD of 20 nM. Thus, Yang’s group developed a colorimetric immunoassay with this strategy to determine the cancer biomarker α-fetoprotein [115]. Cystine can be reduced into cysteine by AA, which in turn facilitated the aggregation of AuNPs [116]. Reversely, Lin’s group constructed an ultrasensitive colorimetric immunosensor for the H7N9 detection virus by employing ALP to catalyze the hydrolysis of AA-P into AA [117]. After binding to the surface of NPs, thiol molecules with positive charges can alter the surface charge distributions of NPs and induce the aggregation of NPs due to the electrostatic interactions.

Peptides can protect NPs from aggregation, and the enzyme/target-induced changes of the electronegativity, configuration, and structure of the peptide may induce the cross-linkage of the dispersed NPs [118]. Jiang’s group developed a colorimetric immunoassay for simultaneous determination of multiple biomarkers (interleukin-6 or IL-6, procalcitonin or PCT, and C-reactive protein or CRP) in a broad and tunable detection range (from pg/mL to g/mL) based on phosphorylated short peptide-mediated controlled aggregation of modification-free AuNPs (Figure 10) [119]. Three types of peptides were designed and chosen in this work. The negatively charged phosphate group near to the arginine (Arg) or lysine (Lys) inhibited the binding of the peptide to AuNPs through Au–N interactions. AuNPs remained dispersed when being incubated with the phosphorylated peptide. With the addition of ALP, the negatively charged phosphate group in the peptide was efficiently removed. The produced positively charged peptide induced the cross-linking aggregation of AuNPs.

Cu(I) derived from the reduction of Cu(II) by AA can catalyze the click reaction between azide and alkyne groups via the Huisgen’s reaction [120]. Based on the Cu(I)-triggered click reaction, AuNPs functionalized, respectively, with azide and alkyne groups can undergo aggregation (Figure 11A) [121]. Based on this concept, Jiang’s group reported a colorimetric immunosensor for the assay of human immunodeficiency virus (HIV) by measuring Cu(II) released from antibodies-labeled copper monoxide nanoparticle (CuO NP) (Figure 11B) [122]. Then, they also employed ALP-catalyzed Cu(I)-triggered azide/alkyne cycloaddition (CuAAC) between different functionalized AuNPs to develop a plasmonic nanosensor for the monitoring of ALP activity and the detection of rabbit antihuman IgG (Figure 11C) [123]. Besides, Ag(I) can exchange the H within the C–H bond of the alkyne to form an alkynyl Ag(I) complex [124]. Long’s group reported a metal-linked immunosorbent assay for the determination of α-fetoprotein, PSA, and C-reactive protein (CRP) [125]. In that paper, AgNPs conjugated with the detection antibody were oxidatively dissolved by H_2_O_2_ to produce millions of silver ions. The released Ag(I) ions induced the aggregation of alkyne-functionalized AuNPs.

### 4.2. Size/Morphology Change of Nanoparticles

Although AuNPs have been employed to detect various targets, they often suffer from strong false positive results (auto-aggregation) and time-consuming modifications of AuNPs. To solve these problems, many researchers have started to design gold or silver nanoparticles-etching or growing sensor platforms based on the shape-dependent property of LSPR. In those works, the biocatalytic reactions of native enzymes, including GOx [31], alcohol dehydrogenase (ADH) [126], and ALP [127,128], can increase or decrease the concentration of substrate or product (H_2_O_2_, NADH, *p*-aminophenol) that can favor the growth or etching of AuNPs to induce the multi-color change. Normally, the absorption and extinction coefficients of NPs with small diameters allow for the colorimetric detection of analytes at the concentration of 10 nM. However, with the gradual enlargement, the colorless NPs solution can turn red along with an appearance of an LSPR absorption band. Meanwhile, the presence of metal NPs can effectively catalyze the reduction and deposition of metal ions on the surface of NPs [129,130]. Since 2004, Willner’s groups have employed enzyme-induced growth or enlargement of AuNPs for some optical detection applications for the determination of small biomolecules [131,132,133], protein [134], and inhibitors of enzymes [135]. 

H_2_O_2_ can reduce Au(III) ions into Au atoms to produce nanoparticles. In 2012, Stevens’s group reported the H_2_O_2_-based growth of AuNPs and developed an ultrasensitive colorimetric immunoassay by the catalase-catalyzed decomposition of H_2_O_2_ (Figure 12A) [34,136]. In this communication, H_2_O_2_ can reduce HAuCl_4_ into quasi-spherical, non-aggregated AuNPs in the absence of the target. After the capture of the analyte and catalase label, H_2_O_2_ molecules are largely consumed by catalase and the kinetics of the NPs’ growth is slowed down, leading to ill-defined and aggregated AuNPs. After that, a variety of enhanced colorimetric immunoassays were developed based on the generation or growth of metal NPs in the presence of HAuCl_4_ or AgNO_3_ [137,138]. For example, Xiong’s group improved the performance of plasmonic ELISA based on the catalase (CAT)-mediated AuNPs’ growth by using silica nanoparticles carrying poly(acrylic acid) (PAA) brushes as the carriers to increase the loading number of the enzymes. The immunosensor exhibits high specificity and sensitivity for *L. monocytogenes* with an LOD of 80 CFU/mL (Figure 12B) [139]. Chen’s group reported the quantitative immunoassay of PSA from attomolar to picomolar levels based on the GOx-catalyzed growth of 5 nm AuNPs, in which the LOD of 93 aM is lower than that of the commercial ELISA (6.3 pM) by at least 4 orders of magnitude (Figure 13C) [31]. Long’s group developed a plasmonic ELISA for hepatitis B surface antigen (HBsAg) and AFP with alcohol dehydrogenase (ADH)-catalyzed AuNPs seed-mediated growth as a colorimetric signal-output strategy (Figure 12D) [126]. The captured conjugates catalyzed the reduction of NAD^+^ to produce NADH. The produced NADH can reduce HAuCl_4_ to result in the enlargement of AuNPs’ seeds and a color change from yellow to purple. Meanwhile, the ALP-mediated growth of Ag NPs was also demonstrated by Liu’s group for sensitive and naked-eye determination of cancer biomarkers in clinical serum samples [140]. However, the immunoassays based on the growth of AuNPs are easily influenced by reductive coexistences in solutions due to the strong oxidation ability of HAuCl_4_. To address this problem, Zhang’s group developed a novel immunoassay strategy for H9N2 AIV detection based on enzyme-induced silver deposition on the surface of AuNPs by using ALP as a signal enzyme (Figure 12E) [128]. Jiang’s group reported the competitive colorimetric detection of interleukin-6 based on catalase-catalyzed hydrolysis of H_2_O_2_ to regulate the deposition of the Ag shell on the surface of AuNPs [141].

The anisotropic metallic NPs with well-defined and controlled shapes, aspect ratios, and high surface energies in the sharp edges of NPs are prone to oxidation and deposition of the metal shell and prefer to develop multicolor immunoassays, such as gold nanorods (AuNRs) and gold nanobipyramids (Au NBPs). The small change in the local refractive index or shape will induce a multi-color change and significant spectral shift in the UV-visible spectra. Based on enzyme-catalyzed reduction of silver ions and the consequent deposition of ultrathin silver shells on gold NRs, several LSPR-based immunoassays have been constructed. For example, Tang’s group reported the colorimetric detection of phosphatase activity (ALP as a model enzyme) and further explored the quantitative determination of rabbit IgG (RIgG) (Figure 13A) [142]. In this work, AA produced from 4-APP by ALP-catalyzed hydrolysis facilitated the reduction of silver ions to generate a silver nanoshell on the surface of AuNRs. The longitudinal LSPR band of AuNRs gradually shifted to a shorter wavelength, and the solution changed from red to orange, yellow, green, cyan, blue, and, further, to violet. The method allowed for the semi-quantitative determination of ALP activity in real-life biological samples by monitoring the change in the color or the longitudinal LSPR peak. Moreover, Gao’s group proposed a signal amplification immunoassay based on the conversion of p-aminophenol phosphate into p-aminophenol by alkaline phosphatase [127]. Very recently, Liu and co-workers constructed a dual-modal split-type photoelectrochemical and colorimetric immunosensor for microcystin-LR (MC-LR) detection by using Au nanobipyramids (Au NBPs) instead of AuNRs [143].

Some oxidants (such as I_2_ [144], H_2_O_2_ [145], TMB^2+^ [15], Cu^2+^ [146]) or reactions (such as the Fenton reaction) [147] can cause the oxidization of Au/Ag into Au(I)/Ag(I) in certain situations and thus lead to a change of the shape of anisotropic NPs to a sphere with a blue-shift of the LSPR absorption. By iodine-mediated etching of AuNRs, Chen’s group proposed a plasmonic ELISA for the sensitive quantification of human IgG in fetal bovine serum (Figure 13B) [144]. In this work, the produced AA through the dephosphorylation of 4-APP by ALP can reduce KIO_3_ into I_2_. Au was then oxidized into Au(I) by the produced I_2_ in the presence of I^−^ and cetyltrimethylammonium. In addition, the anisotropic NPs can be etched into smaller spherical NPs by H_2_O_2_, accompanyied by a blue shift of the LSPR peak and a colorimetric change [148]. Tang’s group reported a plasmonic ELISA for the quantitative assessment of PSA based on antibody-labeled GOx-catalysed oxidation of glucose (Glu) to produce H_2_O_2_ and the triangular Ag nanoprism [145]. Guo’s group suggested that Fenton’s reaction can promote the oxidation of AuNRs by H_2_O_2_ and reported the vivid color display of AuNRs with various types of immunoassays for the visual detection of different targets with the naked eye (Figure 13C) [147]. Lin’s group first demonstrated that TMB^2+^, a typical HRP-catalyzed colorimetric product, can quantitatively and efficiently etch AuNRs. This method was further utilized for semiquantitative detection of disease biomarkers (CEA and PSA) [15]. Chen and co-workers proposed a highly sensitive colorimetric immunoassay with an LOD of 0.15 ng/ml for human IgG based on Cu^2+^-mediated etching of NRs [146]. In this work, Cu^2+^ was released from goat anti-human IgG-labeled CuS NPs by hydrobromic acid.

## 5. Nanomaterials for Loading of Natural or Artificial Enzymes

In conventional immunoassays, the sensitivity and the association constant of substrate–antibody complexes were strictly limited by the low ratio (1:1) of the detection antibody and enzyme label. To address this problem, various nanomaterials exhibiting large surface to volume ratios and rich surface chemistry have been utilized in colorimetric immunoassays as the carriers of target-specific antibodies and enzymatic labels (natural enzymes, nanozymes, and enzyme-mimic molecules). The tagged biomolecules retained their binding and catalysis activity. The surface of a single nanoparticle may accommodate a large number of signaling enzymes and a minimum number of detection antibodies through the rational ratio of the enzyme, antibody, and nanomaterial. Therefore, nanomaterials can significantly improve the performances of bioassays since many enzyme molecules are indirectly bounded for signal amplification, such that even only one nanoparticle can bind to the analyte. Apart from the simple loading properties, nanomaterials can also preserve and enhance the catalytic activity of the encapsulated enzymes against different harsh conditions. For example, MOFs can also preserve and enhance the catalytic activity of the encapsulated enzymes due to the small pore size of MOFs and the coordination interaction between enzymes and MOFs [85,149]. GO can modulate the peroxidase-like property of lysozyme-stabilized Au NCs so that it shows high catalytic activity over a broad pH range, even at neutral pH [150]. 

### 5.1. Zero-Dimensional Nanomaterials

AuNPs as excellent biological signal-amplification tags are one of the most widely used nanocarriers for various kinds of sensors and bioassays [151,152,153,154]. Normally, except for the catalytic properties, MNPs are mainly used to separate sandwich-like immunocomplexes from the reaction mixture and re-disperse immediately after the removal of the magnet for the concentration and purification of analysts. For example, Ambrosi et al. developed an ELISA immunosensor for the detection of cancer antigen 15-3 (CA15-3) antigen using AuNPs as the carriers of the signaling anti-CA15-3-HRP. The immunoassay exhibited a higher sensitivity and shorter assay time in contrast to the classical ELISA (Figure 14A) [153]. Wu’s group designed a comprehensive strategy for the detection of *Salmonella enterica* serovar Typhimurium (STM) based on HRP and detection antibody-modified AuNPs (Figure 14B) [155]. After STM was captured and separated by aptamer-modified magnetic particles, AuNPs carrying a large amount of HRP molecules were added into the solution for colorimetric signal amplification. 

Besides natural enzymes, nanomaterials can also be used to load peroxidase-like organic molecules to regulate the catalytic performances of free molecules. Typically, Xiong’s group employed multi-branched gold nanoflowers (AuNFs) with thiolated amino ligand to load hydrophobic 5,10,15,20-tetraphenyl-21H,23H-porphyrin iron(III) chloride (FeTPPCl) for the inhibition of the catalytic activity of FeTPPCl [156]. When the organic solvent was added, FeTPPCl was released from the hydrophobic surface of AuNFs and its catalytic activity was recovered. Thus, the AuNF@FeTPPCl nanocomposites were used to construct colorimetric biosensors for the detection of fumonisin B1 and hepatitis B surface antigen.

### 5.2. One-Dimensional Nanomaterials

Although AuNPs have increased the sensitivity of immunoassays, the limited size of AuNPs will inevitably hinder further improvements, especially when large biomolecules were assembled on the surface of AuNPs. Alternatively, one- or two- dimensional nanomaterials can provide a larger surface area for the co-assemblage of large recognition elements and signaling enzymes. Carbon nanotubes (CNTs) possess unique electronic, chemical, and mechanical advantages, such as a high surface-to-volume ratio and electrical conductivity. Thus, CNTs are also extremely attractive as the immobilization scaffolds for antibodies and enzymes to prepare sensitive immunosensors [157,158]. Song’s group used multi-walled carbon nanotubes (MWNTs) to load anti-IgG and HRP instead of the conventional anti-IgG-HRP conjugate and demonstrated the application of the MWNTs-based multicomponent probe for colorimetric detection of ataxia telangiectasia mutated (ATM) (Figure 15) [159]. In this study, carboxylated MWNTs were conjugated with anti-IgG and HRP through the amine coupling reaction to amplify the colorimetric signal. The sensitivity of the method was 5000 times higher than a conventional ELISA with an LOD of 0.2 fg/mL or 54 aM (~32 molecules in 1 μL samples). Meanwhile, Surareungchai’s group presented a proof-of-concept ELISA for *S. enterica* serovar Typhimurium detection using single-walled carbon nanotubes (SWCNTs) instead of MWNTs [160]. 

### 5.3. Two-Dimensional (2D) Layered Nanomaterials

After the discovery of graphene, 2D layered nanomaterials, including graphene, layered double hydroxides (LDH), boron nitride (BN), and transition metal dichalcogenides (TMDs), have gained global attraction and have been widely researched in electrochemical, fluorescent, and colorimetric immunoassays because of their good physical and chemical properties. Besides the direct modification of the electrode surface, graphene and its derivatives are always used as signaling labels for loading natural or artificial enzymes in sandwich-like assays immunoassays. For example, Park’s group prepared a novel nanohybrid by integrating catalytically active Pt NPs and MNPs on the surface of GO [161]. Due to the synergistic catalysis and facile mass transfer, the antibody-conjugated nanohybrid catalyzed the quick oxidation of TMB to produce the blue colored TMB^2+^ within 5 min at room temperature. To increase the sensitivity, recently, Dou’s group modified reduced graphene oxide (rGO) with peroxidase-mimic Au@Pt nanoparticles and HRP [162]. In this work, the double signal amplification system was constituted by HRP and Au@Pt.

## 6. Chromogenic Reactions Induced by the Constituent Elements Released from Nanomaterials

Although nanomaterials can drastically improve the sensitivity of immunoassays in the above strategies, they is still limited by some drawbacks. For example, the loading amount of enzymes or antibodies was limited on the surface of nanomaterials, and the conjugation of enzymes or antibodies and blockage proteins significantly reduces the catalytic activity of nanozymes due to the blocking of the catalytic sites on the nanozymes’ surface. To solve those problems, scientists developed some non-enzyme amplification strategies to increase the sensitivity of colorimetric immunoassays. Based on the tremendous advances in nano(bio)technology, we can take advantage of the high-loading capacity of liposomes and the massive institute elements of the detection antibody-conjugated NPs, such as metal NPs, MOFs, and metal oxide or sulfide NPs. For example, once the sandwich-like structure was formed, the NPs were dissolved into the individual metal ions through chemical etching. The released metal ions can catalyze a certain reaction or coordinate with an organic molecule along with the change of the solution color. In particular, Metal ions including Cu(I)/Cu(II) and Fe(II)/(III) can participate in colorimetric reaction through catalysis or coordination. This allowed for the construction of enzyme-free colorimetric immunoassays. For example, Cu(II) can catalyze the decomposition of H_2_O_2_ through the Fenton-like reaction and thus promote the oxidation of TMB by H_2_O_2_. For this consideration, Liu’s group developed a colorimetric immunosensor for Glypican-3 detection employing the antibody-modified CuO NPs. A large number of Cu(II) ions can be released from the captured NPs with HCl solution to promote the oxidation of TMB by H_2_O_2_ [163]. Lai’s group developed a colorimetric immunoassay for CEA detection based on a copper chromogenic reaction, in which Cu(II) released from CuO NPs coordinates with a synthesized chromogenic of 1,2-diphenyl-2-(2-(pyridin-2-yl)hydrazono)ethanone, together with the colorimetric signal change from colorless to red [164]. We have demonstrated that the released Cu(II) from the captured CuO NPs labels can catalyze the oxidation of AA by O_2_, thus depressing the AA-guided formation of colored AuNPs [165]. The colorimetric assay with an LOD of 0.05 ng/mL showed good applicability for the assay of PSA in serum samples. Furthermore, Cu(I) formed by the chemical reduction of Cu(II) can inhibit the enzymatic activity of HRP to catalyze the oxidation of colorless TMB into blue TMB^2+^ [166,167]. Zhang’s group employed the second antibody-labeled Cu-MOFs to develop a colorimetric method for the detection of dibutyl phthalate (DBP) [168]. In this study, tremendous Cu(II) ions were released from Cu-MOFs under HNO_3_, and were subsequently reduced into Cu(I) ions by AA. The produced Cu(I) reacted with HRP to depress the color change by inhibiting the oxidation of TMB.

Liposomes can be prepared from both natural and synthetic lipids consisting of a lipid bilayer phospholipid and an aqueous core [169]. A large amount of biomolecules or signal tags can be encapsulated inside their inner cavity and suddenly released in the presence of surfactants, thus causing a drastic response, such as colorimetric and photoelectrochemical responses [170,171]. For example, in Section 3.1, the cysteine released from liposomes can induce rapid aggregation of AuNPs [110]. Based on this fact, Tang et al. reported a signal amplification strategy for colorimetric immunoassay of streptomycin (STR) using glucose-trapped liposome [172]. Upon addition of Triton X-100, glucose was released and oxidized into gluconic acid and H_2_O_2_ by GOx. The produced H_2_O_2_ can oxidize Fe(II) into (III), thus inhibiting the formation of the Fe(II)-phenanthroline complex and inducing the color change from orange-red to light-yellow. Moreover, Luo’s group demonstrated that HRP encapsulated into liposome can be released by TMB to catalyze the H_2_O_2_-mediated colorimetric oxidation of TMB [173]. 

Organic compounds (e.g., pH-sensitive chromatic dyes) can be absorbed on the surface or embedded in the interior of nanocarriers through non-covalent bonding or encapsulation interactions. The compounds can be readily released to the aqueous solution under a certain stimulation, thus releasing in an obvious allochroic response. This signal strategy has also been used in developing colorimetric immunoassays. Typically, Li’s group designed a pH indicator-linked immunoassay using carboxylic functionalized carbon nitride (cC_3_N_4_) nanosheets to load phenolphthalein (pH indicator) as the signal labels [174]. The cargo phenolphthalein was released from cC_3_N_4_ into anions to show a pink color under an alkali solution. This proposed method offered a visible signal amplification approach for simple, low-cost, and stable colorimetric detection of cancer biomarkers. Based on the same strategy, this group developed a dual-modal colorimetric and fluorescent immunosensor for the detection of cardiac troponin I (cTnI) (Figure 16A) [175]. In this system, three-dimensional MoS_2_ nanoflowers (3D-MoS_2_ NFs) were used to immobilize curcumin (CUR) by the hydrophobic interactions and quench the fluorescence through fluorescence resonance energy transfer. Under the stimulus of OH^−^, hydrophobic CUR was converted into hydrophilic ion and released from the MoS_2_ NFs, thus producing a distinct color change and a “signal-on” fluorescence signal simultaneously. Additionally, Li’s group also developed an enzyme-free titer plate-based colorimetric assay prostate specific antigen (PSA) utilizing functionalized mesoporous silica nanoparticles (MSNs) to incorporate pH-sensitive thymolphthalein (TP) (Figure 16B) [176]. They functionalized MSN poles with phenyltrimethyloxysilane to tightly entrap TP through a π–π stacking interaction. The TP-containing MSNs were covered with polyethylenimine (PEI) to attach the secondary anti-PSA antibody. After the formation of the immunocomplex, the entrapped TPs were released from the pores under an alkaline solution. Meanwhile, Tang et al. reported the detection of AFP by employing a TP-modified metal-polydopamine framework (MPDA@TP) for the signal amplification (Figure 16C) [177]. The hollow-structured MPDA@TP was modified with TP by the π-stacking interaction using zeolitic imidazolate frameworks (ZIF-67) as the templates.

## 7. Conclusions

In summary, we reviewed the recent achievements in nanomaterials-based colorimetric immunoassays for the sensitive detection of various analytes. The integration of various nanomaterials to colorimetric immunoassays has significantly improved the analytical performances with the advantages of nanomaterials, including tunable chemical and physical properties, enzyme-mimic abilities, and facile modification. Thus, nanomaterials-based colorimetric immunoassays are promising candidates to conventional colorimetric enzyme-linked immunoassays. With the integration of nanotechnology, analytical methodology, and biotechnology, the potential capacities and applications of immunoassays are highly desired, especially in resource-poor setting areas. However, despite the significant advances in the novelty of the designed methods and the significant improvement of the sensitivity and selectivity, there are still several challenges in their implementation in biomedical fields and the translation from lab investigation to clinical diagnosis. At present, a general disadvantage for all the nanomaterial-based immunoassays is that their reproducibility and stability are less than traditional assays due to the experimental and systemic factors. This issue should not be an obstacle to the construction of methods for the use of nanomaterials, as the development of industrial technology will ultimately ensure the standardization of nanomaterials’ production. The preparation and utilization of immunoassays or commercial kits require proper washing and surface blocking steps and professional training. This point is important to reduce the false positive or negative response. Typically, non-specific adsorption of interfering agents often occurs on the surface of nanomaterials of biological samples because of the large specific surface area of nanomaterials. Although antifouling agents, such as polyethylene glycol (PEG) and bovine serum albumin, can resist surface contamination, there are still many challenges. For instance, PEG can suffer from auto-oxidation in biological samples, so it cannot be kept for a long time in commercial diagnostic kits. We believe that special attention should be given to improve the specificity and reduce the inaccuracy of the reviewed methods in the future.

## Figures and Tables

**Figure 1 nanomaterials-09-00316-f001:**
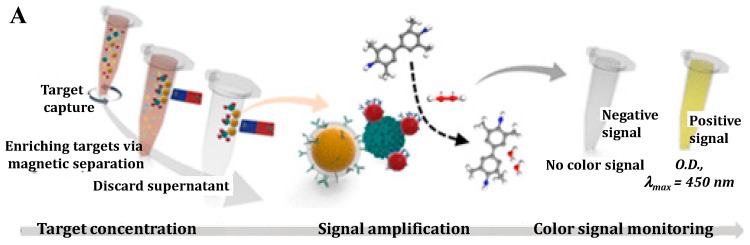
(**A**) Schematic illustration of the magnetic nanobead-based nano(e)zyme-linked immunosorbent assay (MagLISA). Reproduced with permission from [47]. Copyright American Chemical Society, 2018. (**B**) Schematic representation of the plasmonic nanosenor, in which the target molecule is anchored to the substrate by capture antibodies and recognized by other antibodies labeled with gold nanoclusters (AuNCs). Reproduced with permission from [50]. Copyright American Chemical Society, 2016.

**Figure 2 nanomaterials-09-00316-f002:**
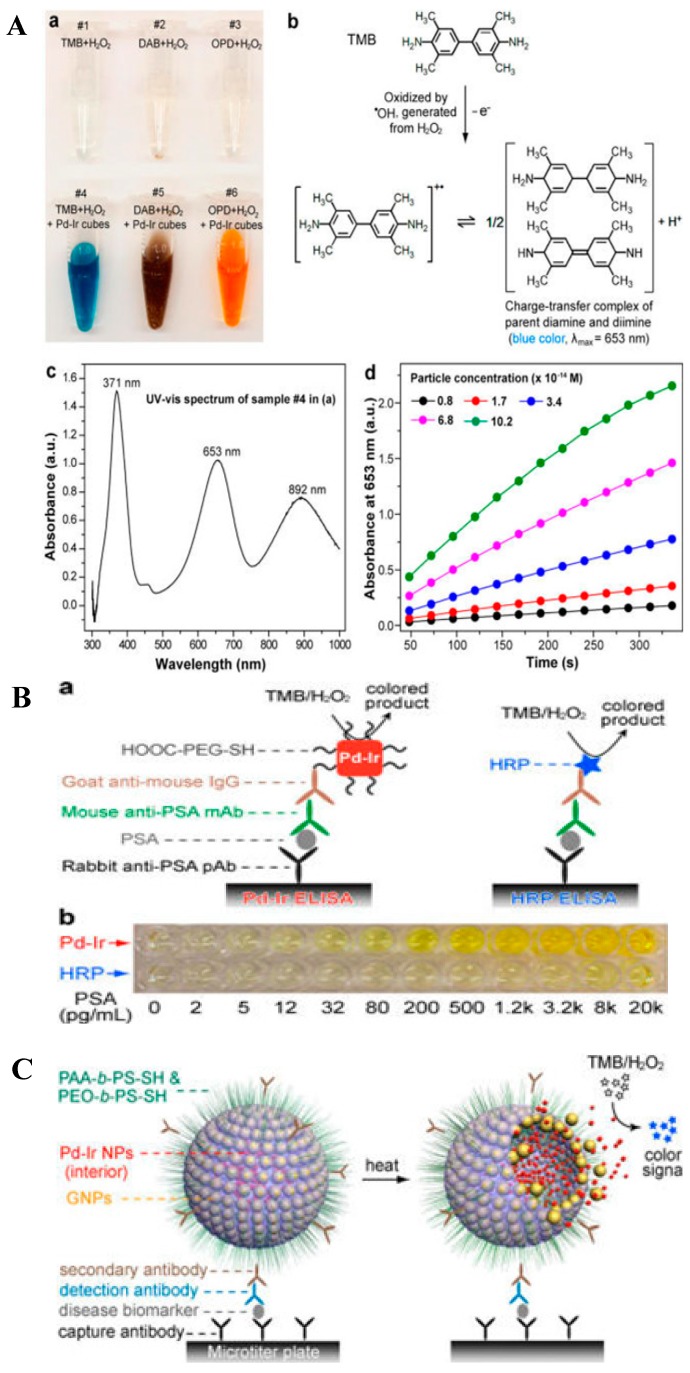
(**A**) Peroxidase-like activity of Pd-Ir cubes. Reproduced with permission from [54]. Copyright American Chemical Society, 2015. (**B**) Detection of prostate surface antigen (PSA) with Pd-Ir cubes-based enzyme-linked immunosorbent assay (Pd-Ir ELISA) and conventional horseradish peroxidase-based ELISA (HRP ELISA). Reproduced with permission from [54]. Copyright American Chemical Society, 2015. (**C**) Schematic illustration of the utilization of Pd−Ir NPs@GVs based ELISA for the detection of disease biomarkers. Reproduced with permission from [55]. Copyright American Chemical Society, 2017.

**Figure 3 nanomaterials-09-00316-f003:**
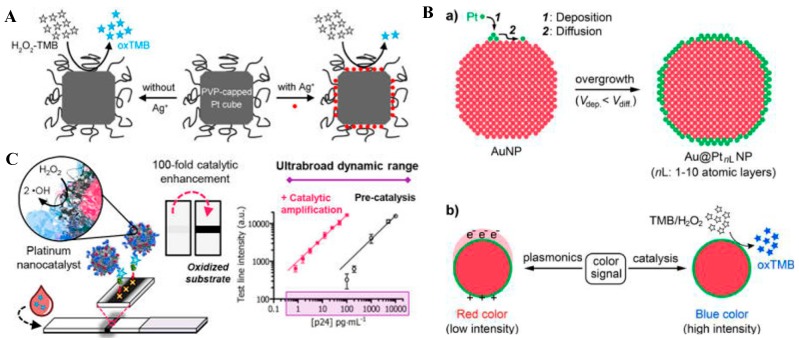
(**A**) Poly-(vinylpyrrolidone) (PVP)-capped Pt cubes for the colorimetric detection of Ag^+^ ions. Reproduced with permission from [58]. Copyright American Chemical Society, 2017. (**B**) Schematics showing (a) the fabrication of Au@Pt NPs in which Pt atoms are deposited onto an AuNP to form a conformal, thin Pt shell with thicknesses of 1–10 atomic layers and (b) two types of color signal generated from Au@Pt*_n_*_L_ NPs under different mechanisms. Reproduced with permission from [60]. Copyright American Chemical Society, 2017. (**C**) Scheme showing amplified lateral flow immunoassay (LFIA), where functionalized Pt nanocatalysts and biotinylated nanobody fragments are mixed with a plasma or serum sample. Reproduced with permission from [61]. Copyright American Chemical Society, 2018.

**Figure 4 nanomaterials-09-00316-f004:**
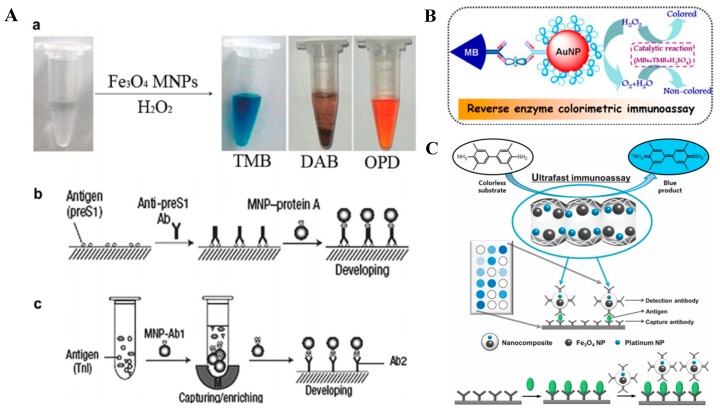
(**A**) The peroxidase-like activity of magnetic nanoparticles (MNPs) and two immunoassays based on the peroxidase activity of MNPs. Reproduced with permission from [62]. Copyright The Nature Publishing Group, 2007. (**B**) Magnetocontrolled enzyme-mediated reverse colorimetric immunosensing strategy. Reproduced with permission from [64]. Copyright American Chemical Society, 2013. (**C**) Immunoassay based on a nanocomposite entrapping both MNPs and Pt NPs in mesoporous carbon. Reproduced with permission from [65]. Copyright John Wiley and Sons, 2014.

**Figure 5 nanomaterials-09-00316-f005:**
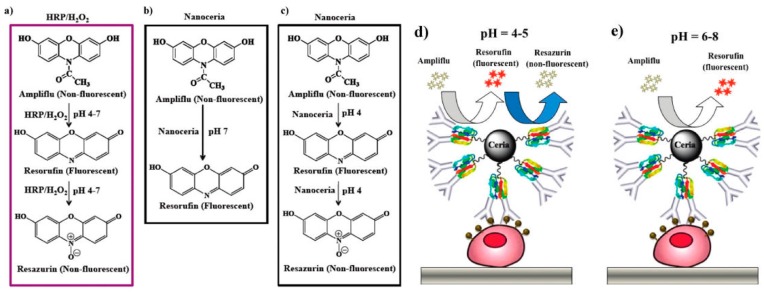
Schematic showing the HRP/H_2_O_2_ and nanoceria mediated oxidation of ampliflu. (**a**) In the pH range of 4–7, HRP/H_2_O_2_ oxidizes ampliflu to a nonfluorescent final product (resazurin); (**b**) in contrast, nanoceria oxidizes ampliflu to the intermediate oxidation fluorescent product (resorufin) at pH 7; (**c**) while at or below pH 5.0, nanoceria yields the terminal oxidized nonfluorescent product, resazurin; (**d**,**e**) the ability of nanoceria to oxidize ampliflu to a stable fluorescent product in the pH range of 6–8 will facilitate its use in ELISA without the use of H_2_O_2_. Reproduced with permission from [70]. Copyright American Chemical Society, 2011.

**Figure 6 nanomaterials-09-00316-f006:**
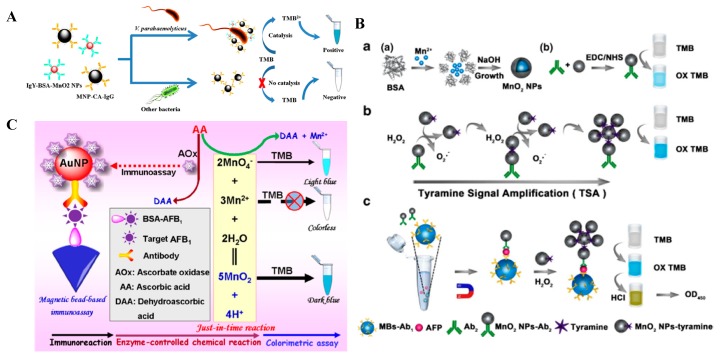
(**A**) Schematic diagram for the immunomagnetic capture and colorimetric detection of *V. parahaemolyticus*. Reproduced with permission from [75]. Copyright Springer Nature, 2017. (**B**) The principles and procedures of the MnO_2_ NPs based immunosensor for α-fetoprotein (AFP) detection. (a) Schematic procedures for preparation of BSA-MnO_2_ NPs and Ab_2_-MnO_2_ NPs conjugate; (b) Schematic illustration for the tyramine signal amplification (TSA) system; (c) procedures for AFP detection through the MnO_2_ NPs-based immunosensor combined with magnetic separation. Reproduced with permission from [76]. Copyright Springer Nature, 2017. (**C**) Schematic illustration of the conventional chemical reaction for an unconventional application in the magnetically responsive colorimetric immunoassay using enzyme-responsive just-in-time generation of MnO_2_ nanocatalyst. Reproduced with permission from [77]. Copyright Springer Nature, 2018.

**Figure 7 nanomaterials-09-00316-f007:**
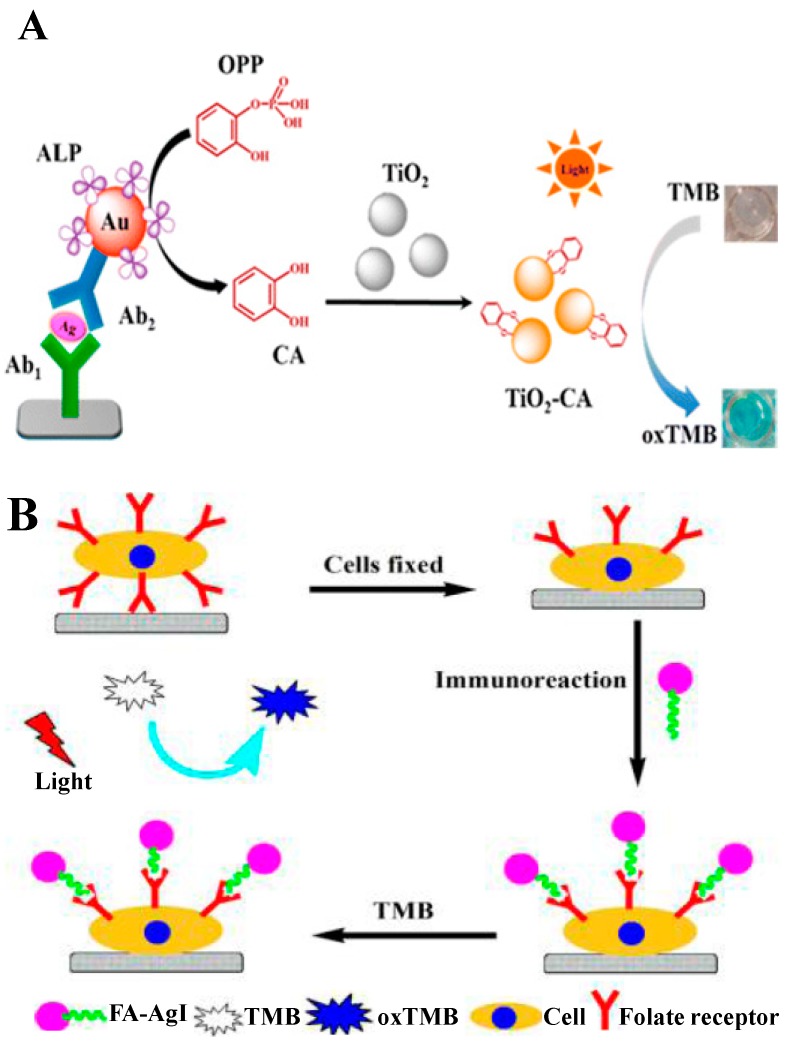
(**A**) Proposed immunodetection process for mouse IgG by coupling the cascade reaction of alkaline phosphatase (ALP) and the enzymatically in situ generated photoresponsive nanozyme of TiO_2_-CA. Reproduced with permission from [80]. Copyright American Chemical Society, 2017. (**B**) Proposed detection process using FA-CS-AgI. Reproduced with permission from [83]. Copyright American Chemical Society, 2017.

**Figure 8 nanomaterials-09-00316-f008:**
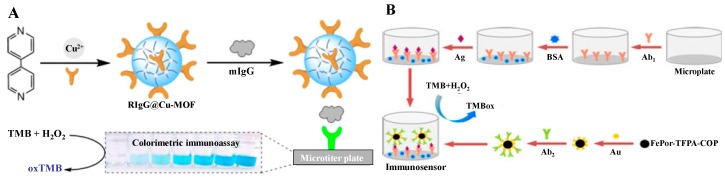
(**A**) Illustration of the colorimetric immunoassay of mIgG based on RIgG@Cu-MOF as a detection antibody. Reproduced with permission from [88]. Copyright American Chemical Society, 2018. (**B**) Illustration of the fabrication of FePor-TFPA-COP-based colorimetric immunoassay. Reproduced with permission from [93]. Copyright American Chemical Society, 2017.

**Figure 9 nanomaterials-09-00316-f009:**
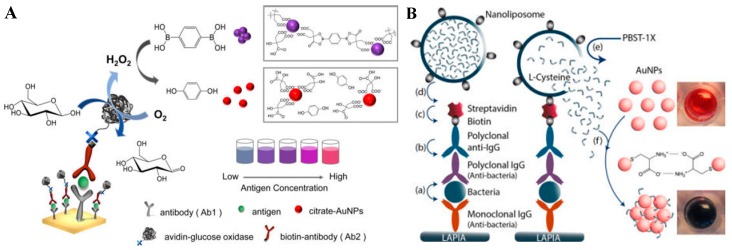
(**A**) Naked-eye readout of plasmonic immunoassays. Detection of target protein via the combination of sandwich immunoassay, avidin−biotin interaction, glucose oxidase (GOx)-mediated oxidation of glucose, H_2_O_2_-induced oxidation of benzene-1,4-diboronic acid (BDBA), and BDBA-triggered aggregation of citrate-capped AuNPs. Reproduced with permission from [108]. Copyright American Chemical Society, 2016. (**B**) Schematic of the liposome-amplified plasmonic immunoassay. Reproduced with permission from [111]. Copyright American Chemical Society, 2015. (**C**) The AChE-catalyzed hydrolysis reaction for the colorimetric detection of enterovirus 71 (EV71). Reproduced with permission from [112]. Copyright John Wiley and Sons, 2013. (**D**) Plasmonic immunoassay based on HRP mediated modulation of AuNPs that enables naked-eye readout. Reproduced with permission from [114]. Copyright American Chemical Society, 2015.

**Figure 10 nanomaterials-09-00316-f010:**
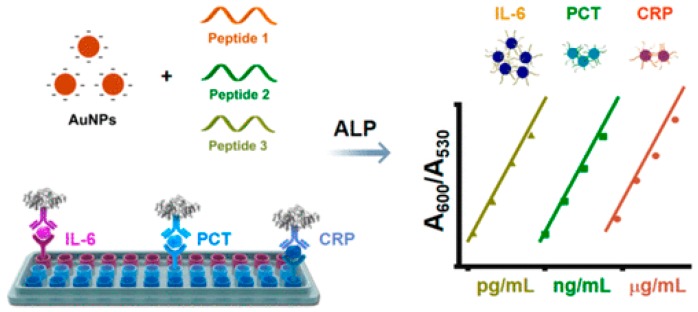
Principle of peptide-ALP-AuNPs immunoassay for simultaneous detection of the inflammatory markers (IL-6, PCT, and CRP). Reproduced with permission from [119]. Copyright American Chemical Society, 2018.

**Figure 11 nanomaterials-09-00316-f011:**
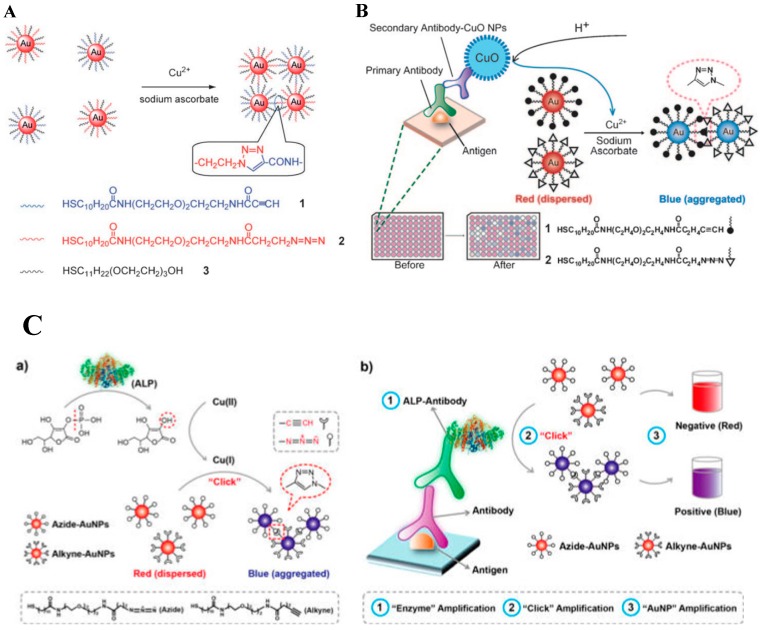
(**A**) The detection of Cu^2+^ ions using click chemistry between two types of gold NPs, each modified with thiols terminated in an alkyne (1) or an azide (2) functional group. Reproduced with permission from [121]. Copyright John Wiley and Sons, 2008. (**B**) Copper-mediated amplification allows a readout of the immunoassay by the naked eye based on CuO-labeled antibody and click chemistry. Reproduced with permission from [122]. Copyright John Wiley and Sons, 2011. (**C**) Plasmonic nanosensor based on ALP-triggered CuAAC between azide- and alkyne-functionalized AuNPs and a naked-eye readout of plasmonic immunoassays based on ALP-triggered CuAAC through three-round amplification. Reproduced with permission from [123]. Copyright American Chemical Society, 2014.

**Figure 12 nanomaterials-09-00316-f012:**
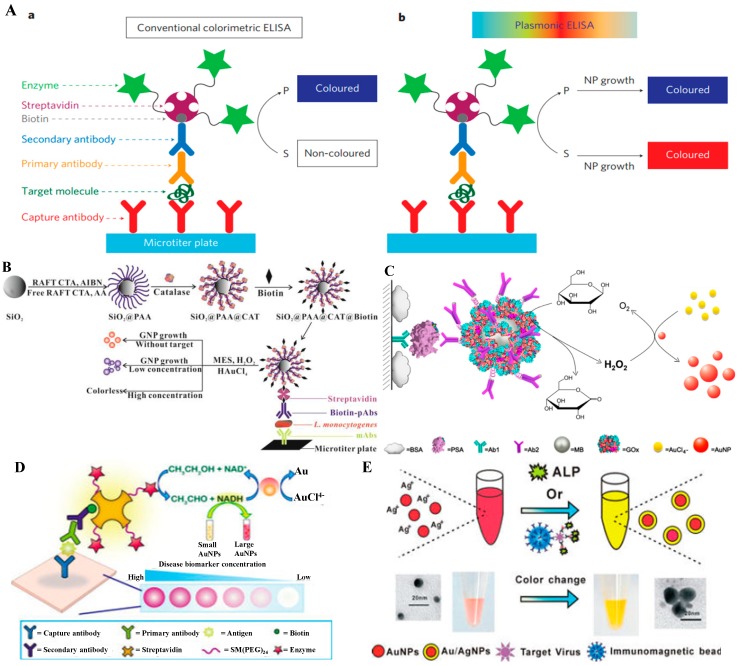
(**A**) Schematic representation of the sandwich ELISA format used here and two possible signal generation mechanisms. Reproduced with permission from [34]. Copyright The Nature Publishing Group, 2012. (**B**) Schematic of the proposed quantitative immunoassay based on SiO2@PAA@CAT-catalyzed growth of AuNPs. Reproduced with permission from [139]. Copyright American Chemical Society, 2015. (**C**) Schematic diagram of the quantitative immunoassay based on glucose oxidase (GOx)-catalyzed growth of gold nanoparticles. Reproduced with permission from [31]. Copyright American Chemical Society, 2014. (**D**) Schematic representation of the sandwich plasmonic ELISA and the signal-generation method. Reproduced with permission from [126]. Copyright American Chemical Society, 2015. (**E**) Illustration of the protocol for the colorimetric magnetoimmunoassay of H9N2 AIV. Reproduced with permission from [128]. Copyright American Chemical Society, 2014.

**Figure 13 nanomaterials-09-00316-f013:**
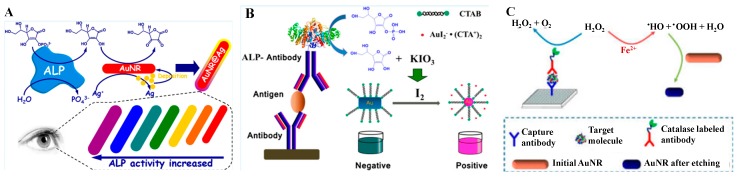
(**A**) Schematic illustration of the high-resolution colorimetric assay for sensitive visual readout of phosphatase activity based on gold/silver core/shell nanorod. Reproduced with permission from [142]. Copyright American Chemical Society, 2014. (**B**) Schematic illustration for visual plasmonic ELISA based on ALP-triggered etching of AuNRs. Reproduced with permission from [144]. Copyright American Chemical Society, 2015. (**C**) Principle of the proposed naked-eye semiquantitative ELISA for visual quantification of proteins. Reproduced with permission from [147]. Copyright American Chemical Society, 2015.

**Figure 14 nanomaterials-09-00316-f014:**
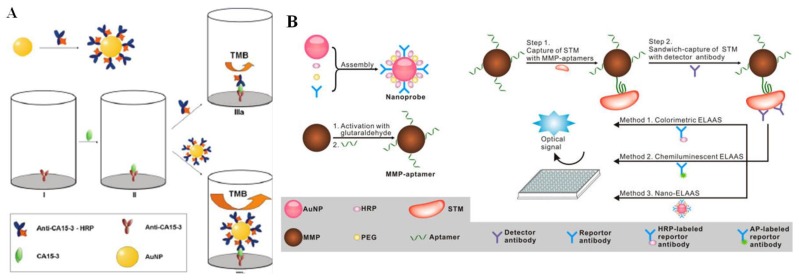
(**A**) Schematic (not in scale) of the preparation of the complex Au-anti-CA15-3-HRP and the sandwich-type ELISA procedure without (IIIa) and with (IIIb) the application of AuNPs as the signal enhancer. Reproduced with permission from [153]. Copyright American Chemical Society, 2010. (**B**) Schematic illustration of serovar Typhimurium (STM) detection based on HRP and detection antibody-modified AuNPs. Reproduced with permission from [155]. Copyright American Chemical Society, 2014.

**Figure 15 nanomaterials-09-00316-f015:**
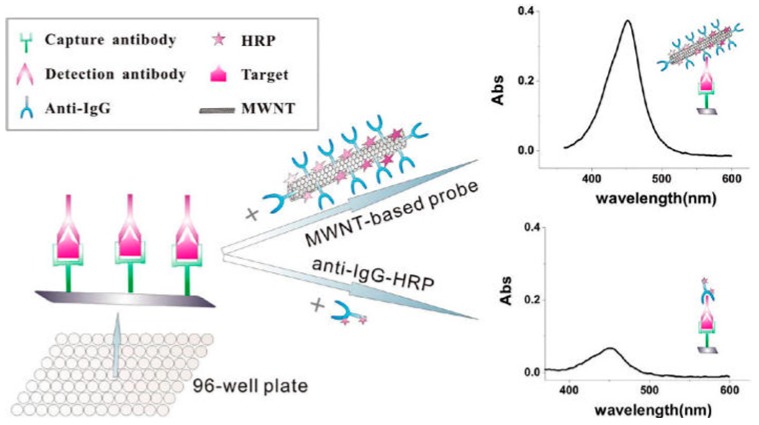
Scheme of the colorimetric strategy for the detection of ataxia telangiectasia mutated (ATM) using an MWNT-based probe compared with conventional ELISA. Reproduced with permission from [159]. Copyright American Chemical Society, 2011.

**Figure 16 nanomaterials-09-00316-f016:**
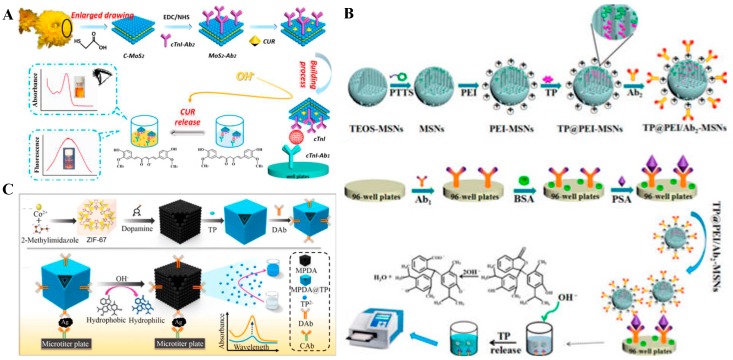
(**A**) Diagram of the drug delivery system inspired enzyme-linked immunosorbent assay (DDS-ELISA) for the detection of cTnI in 96-microwell plates. Reproduced with permission from [175]. Copyright American Chemical Society, 2018. (**B**) Synthesis and derivatization of TP@PEI/Ab_2_-MSNs and steps of the enzyme-free immunosorbent assay of PSA using TP@PEI/Ab_2_-MSNs for amplified colorimetric detection in a 96-well plate. Reproduced with permission from [176]. Copyright American Chemical Society, 2018. (**C**) Schematic representation of MPDA@TP-linked immunsorbent assay (MLISA) for α-fetoprotein (AFP) on anti-AFP capture antibody (CAb)-modified microplate by using anti-AFP detection antibody (DAb)-labeled MPDA@TP with a sandwich-type immunoreaction mode. Reproduced with permission from [177]. Copyright American Chemical Society, 2018.

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
