# Peer review of "Nanomaterials-Based Colorimetric Immunoassays"

_nanomaterials, 2019, doi:10.3390/nano9030316_

Round 1
Reviewer 1 Report
Nanomaterials-Based Colorimetric Immunoassays
In this review, the authors discussed in general the recent advances in colorimetric immunoassay utilizing nanomaterials, in which they are categorized in two as nanozymes or substrates. This manuscript is well-written in describing the trend in the development of the colorimetric nanosensor platform. However, several points should be considered.
1. In introducing this manuscript, the authors stated
“However, the sensitivity of ELISA is strictly limited by the number and the activity of enzyme molecule in the catalytic reaction.”
As it is a limitation of the conjugated enzymes on the immunoassay complex system, this could be applied as well to the nanozyme-based system. Although in the passage of this manuscript, several strategies have been mentioned to overcome this limitation, this point is not highlighted, or rather, the authors did not discuss it as an encouraging point of this review. Please reconsider this.
2. In this review, the authors concluded the challenge of transferring this technique from lab investigation to on-site application or commercial kits, however, the authors are recommended to not only mentioning the methodology in the discussion. As it is a good overview on the current trend through various examples, the challenge and general reason on the need for further enhancement technique on using nanomaterials for colorimetric bioassay should be addressed to boost the research on the corresponding field of this review.
3. Furthermore, as this is a review on the colorimetric immunoassay using nanomaterials, and in the conclusion, the authors show the real problem on how to implement this system because of efforts to reduce false positive and negative response. Please elaborate this to the discussion.
4. In gold nanozyme section, the authors discuss several application of this system, however, there is an overlapping interest in the mentioned reference, between MNP-enrichment technique (discussed in magnetic nanoparticles) and catalytic activity of Au NPs (main interest). As this review focuses on methodology idea on using nanomaterials and recent advance, please consider emphasizing the main interest in suitable section.
5. As following the comment on point (5), this following publication can be a good insight for this manuscript regarding enhancement strategy for utilizing gold nanozyme in the colorimetric platform.
Biosensors and Bioelectronics 126 (2019): 425-432
This manuscript has a good summary of the wide methodologies related to nanomaterials, which will be helpful in giving broad insight on the current progress of colorimetric immunoassay nanosensor platform.
Author Response
We thank the reviewer for his/her positive comments: “In this review, the authors discussed in general the recent advances in colorimetric immunoassay utilizing nanomaterials, in which they are categorized in two as nanozymes or substrates. This manuscript is well-written in describing the trend in the development of the colorimetric nanosensor platform. However, several points should be considered.”
Comment 1: “In introducing this manuscript, the authors stated “However, the sensitivity of ELISA is strictly limited by the number and the activity of enzyme molecule in the catalytic reaction.” As it is a limitation of the conjugated enzymes on the immunoassay complex system, this could be applied as well to the nanozyme-based system. Although in the passage of this manuscript, several strategies have been mentioned to overcome this limitation, this point is not highlighted, or rather, the authors did not discuss it as an encouraging point of this review. Please reconsider this.”
Response: We have rewritten the sentences as follows: “However, the traditional ELISA still has many shortcomings in the aspects of limited sensitivity, complex operation, high cost and high reagent consumption [4]. Another popular immunoassay method, transverse flow analysis (LFA), is also widely used as another traditional colorimetric immunoassay method. Gold nanoparticles (AuNPs)-related LFA is an early example of nanomaterial-related colorimetric immunoassay. Because LFA is very simple in operation and reading, it shows obvious advantages in point-of-care testing. However, the sensitivity of LFA is usually not comparable to that of ELISA [5].”
Comment 2: “In this review, the authors concluded the challenge of transferring this technique from lab investigation to on-site application or commercial kits, however, the authors are recommended to not only mentioning the methodology in the discussion. As it is a good overview on the current trend through various examples, the challenge and general reason on the need for further enhancement technique on using nanomaterials for colorimetric bioassay should be addressed to boost the research on the corresponding field of this review.”
Response: We have added the following sentences in the conclusion: “At present, a general disadvantage for all the nanomaterial-based immunoassays is that their reproducibility and stability are less than traditional assays due to the experimental and systemic factors. This issue should not be an obstacle to the construction of methods for the use of nanomaterials, as the development of industrial technology will ultimately ensure the standardization of nanomaterials production.”
Comment 3: “Furthermore, as this is a review on the colorimetric immunoassay using nanomaterials, and in the conclusion, the authors show the real problem on how to implement this system because of efforts to reduce false positive and negative response. Please elaborate this to the discussion.”
Response: We have added the following sentences in the conclusion: “At present, a general disadvantage for all the nanomaterial-based immunoassays is that their reproducibility and stability are less than traditional assays due to the experimental and systemic factors. This issue should not be an obstacle to the construction of methods for the use of nanomaterials, as the development of industrial technology will ultimately ensure the standardization of nanomaterials production.” and “Typically, non-specific adsorption of interfering agents often occurs on the surface of nanomaterials of biological samples because of the large specific surface area of nanomaterials. Although antifouling agents such as polyethylene glycol (PEG) and bovine serum albumin can resist surface contamination, there are still many challenges. For instance, PEG can suffer from auto-oxidation in biological samples, so it can not be kept for a long time in commercial diagnostic kits. We believe that special attention should be paid to improve the specificity and reduce the inaccuracy of the reviewed methods in the future.”
Comment 4: “In gold nanozyme section, the authors discuss several application of this system, however, there is an overlapping interest in the mentioned reference, between MNP-enrichment technique (discussed in magnetic nanoparticles) and catalytic activity of Au NPs (main interest). As this review focuses on methodology idea on using nanomaterials and recent advance, please consider emphasizing the main interest in suitable section.”
Response: Magnetic nanoparticles (MNPs) are particularly useful for a wide range of biomedical, environmental and catalytic application. They can be conjugated with enzymes, DNA, peptides or antibodies for further functionalization. In gold nanozyme section, MNPs were used as the carriers for the capture and separation of targets and gold nanozyme. In the metal oxide-based nanozymes section, Fe3O4 MNPs exhibiting an intrinsic peroxidase-like activity similar to the natural peroxidases were used for the signal output. We have rewritten the two sections and emphasized the main interest in suitable section.
Comment 5: “As following the comment on point (5), this following publication can be a good insight for this manuscript regarding enhancement strategy for utilizing gold nanozyme in the colorimetric platform. Biosensors and Bioelectronics 126 (2019): 425-432”
Response: We have cited the reference and added the comment on Page 5.
Comment 6: “This manuscript has a good summary of the wide methodologies related to nanomaterials, which will be helpful in giving broad insight on the current progress of colorimetric immunoassay nanosensor platform.”
Response: We thank the reviewer again for his/her positive comment.
Reviewer 2 Report
This review article is well written and easy to follow. It provides a thorough and complete view on the different approaches used so far in nanoparticle-based immunodetectiond. There are only a few typos that should be corrected prior to publication. I recommend to add a list of abbreviations.
Author Response
Comments: “This review article is well written and easy to follow. It provides a thorough and complete view on the different approaches used so far in nanoparticle-based immunodetectiond. There are only a few typos that should be corrected prior to publication. I recommend to add a list of abbreviations.”
Response: We thank the reviewer for his/her positive comments. We have checked the main text carefully and added the full names of all the abbreviations.
Reviewer 3 Report
In this review, the authors are discussing the recent achievements for developing the colorimetric immunoassay using nanomaterials. This manuscript is very good written in describing the trend in the development of the colorimetric immunosensors as well as the transfer of these efforts for the commercial application. However, I saw this review illustrates the recent advances in the development of the colorimetric immunosensors based on using nanometals but the authors in this review didn’t refer to the efforts for developing highly sensitive colorimetric immunosensors utilizing polymeric nanomaterials such as electrospun nanofibers that will open new horizon in the development of immunosensor development whether colorimetric or electrochemical. I recommend adding another section about the polymeric nanomaterials or change the title of the review to be “Nanometals-based colorimetric immunoassays”.
Herein some article for employing the nanofibers for development the immunoassays:
Materials Science and Engineering C 58 (2016) 586–594.
Biosensors and Bioelectronics 38 (2012) 209–214.
Analytical Chemistry 2015, 87 (23), 11863–11870.
Biosensors and Bioelectronics 69 (2015) 257–264
Sensors and Actuators B 205 (2014) 50–60.
Materials 9 (2016) 47.
Biosensors and Bioelectronics 67 (2015) 560–569.
Author Response
Comments: “In this review, the authors are discussing the recent achievements for developing the colorimetric immunoassay using nanomaterials. This manuscript is very good written in describing the trend in the development of the colorimetric immunosensors as well as the transfer of these efforts for the commercial application. However, I saw this review illustrates the recent advances in the development of the colorimetric immunosensors based on using nanometals but the authors in this review didn’t refer to the efforts for developing highly sensitive colorimetric immunosensors utilizing polymeric nanomaterials such as electrospun nanofibers that will open new horizon in the development of immunosensor development whether colorimetric or electrochemical. I recommend adding another section about the polymeric nanomaterials or change the title of the review to be “Nanometals-based colorimetric immunoassays”. Herein some article for employing the nanofibers for development the immunoassays: Materials Science and Engineering C 58 (2016) 586–594. Biosensors and Bioelectronics 38 (2012) 209–214. Analytical Chemistry 2015, 87 (23), 11863–11870. Biosensors and Bioelectronics 69 (2015) 257–264. Sensors and Actuators B 205 (2014) 50–60. Materials 9 (2016) 47. Biosensors and Bioelectronics 67 (2015) 560–569.”
Response: We thank the reviewer for his/her positive comments. This review summarized the recent advances in the development of colorimetric immunosensors with nanomaterials as signal labels. Polymeric nanoparticles have been used to enhance the capture efficiency of sensor surfaces. However, no colorimetric immunosensors were reported with the polymeric nanoparticles as the signal labels. We have cited the references and added the following sentences in the revised manuscript: “Nanomaterials such as polymeric nanoparticles or nanofibers can enhance the capture efficiency of sensor surfaces due to the large surface-to-volume ratios [8,9,10,11,12,13]. Efficient immobilization of antibodies on nanoscale surface for enhancing capture of target is not summarized in this review. Due to the explosion of academic papers related to this extremely wide research field, we may undoubtedly miss many important findings. We sincerely apologize to the authors for their interesting works that are overlooked in this review.”